# DrugnomeAI is an ensemble machine-learning framework for predicting druggability of candidate drug targets

Arwa Raies [1], Ewa Tulodziecka[1], James Stainer[1], Lawrence Middleton[1], Ryan S. Dhindsa [2,3], Pamela Hill[4], Ola Engkvist [5], Andrew R. Harper[1], Slavé Petrovski[1,6] & Dimitrios Vitsios [1✉]

The druggability of targets is a crucial consideration in drug target selection. Here, we adopt a stochastic semi-supervised ML framework to develop DrugnomeAI, which estimates the druggability likelihood for every protein-coding gene in the human exome. DrugnomeAI integrates gene-level properties from 15 sources resulting in 324 features. The tool generates exome-wide predictions based on labelled sets of known drug targets (median AUC: 0.97), highlighting features from protein-protein interaction networks as top predictors. DrugnomeAI provides generic as well as specialised models stratified by disease type or drug therapeutic modality. The top-ranking DrugnomeAI genes were significantly enriched for genes previously selected for clinical development programs ($p$ value $< 1 \times 10^{-308}$) and for genes achieving genome-wide significance in phenome-wide association studies of 450 K UK Biobank exomes for binary ($p$ value $= 1.7 \times 10^{-5}$) and quantitative traits ($p$ value $= 1.6 \times 10^{-7}$). We accompany our method with a web application (http://drugnomeai.public.cgr.astrazeneca.com) to visualise the druggability predictions and the key features that define gene druggability, per disease type and modality.

[1] Centre for Genomics Research, Discovery Sciences, BioPharmaceuticals R&D, AstraZeneca, Cambridge, UK. [2] Centre for Genomics Research, Discovery Sciences, BioPharmaceuticals R&D, AstraZeneca, Waltham, MA, USA. [3] Department of Molecular and Human Genetics, Baylor College of Medicine, Houston, USA. [4] Emerging Innovations, Discovery Sciences, BioPharmaceuticals R&D, AstraZeneca, Waltham, MA, USA. [5] Molecular AI, Discovery Sciences, R&D, AstraZeneca, Gothenburg, Sweden. [6] Department of Medicine, University of Melbourne, Austin Health, Melbourne, VIC, Australia. ✉email: dimitrios.vitsios@astrazeneca.com

Druggability (also known as tractability) is an important concept in drug discovery that influences target identification and may impact the clinical development success rate. The concept of the "druggable genome" was first introduced in 2002[1], and was then defined as the ability of a protein to bind a modulator and provide a desired therapeutic effect. Estimates suggest ~22% of genes in the human genome are druggable by conventional small molecule and monoclonal antibodies[2], with only half of these demonstrating disease associations[3]. Whilst prior druggability predictions were reliant on targets of approved compounds and broad gene classifications, it is plausible that with a wealth of large-scale gene- and systems biology-level data, such as population-based intolerance metrics, tissue expression data and protein structures, our ability to predict druggability may have improved. This work aims to both expand our understanding of druggability, by illuminating gene properties that influence their binding affinity, and facilitate the target identification process, by highlighting genes that have similar druggability profiles to successful drug targets.

While the focus of this study is on druggability, it should be noted that druggability differs from the related concept of ligandability. The latter refers to the ability to develop a modulator (e.g. a small molecule) that can bind to a protein[4], while druggability focuses on the ability to elicit a therapeutic effect because of activation or inhibition of the gene by a modulator and the modulator's ability to reach the protein (e.g. a modulator passing the cell membrane in the case of intracellular targets). Therefore, some genes may be ligandable but not necessarily druggable. When there are two equally ligandable targets, the goal is to prioritise the one that is more likely to be druggable.

Predicting druggable genes using standard machine learning (ML) approaches has been historically challenging due to the relatively small number of known druggable targets, high data imbalance and lack of reliable negative samples[5,6]. To address these issues, we developed DrugnomeAI, a ML framework for ranking genes according to their predicted druggability scores. This is based on mantis-ml[7], an ML tool for gene prioritisation, which addresses the aforementioned challenges by incorporating a stochastic semi-supervised learning approach on positive-unlabelled data. We integrated multiple data sources into the framework resulting to 324 generic and druggability-specific gene-level features, including protein-protein interaction and pathway-based data, to elucidate the druggability problem at a systems biology level. We developed multiple druggability models to produce exome-wide druggability profiles with varying levels of stringency, based on the underlying evidence provided by different historical labelled datasets of successful drug targets. The end goal is to provide a holistic and unbiased view of the druggability profile of each gene, as it is captured by multiple models, each contributing to a different aspect of their druggability potential.

Several resources have been created for assessing target's druggability. Some databases, such as Open Targets[8] and TractaViewer[9] provide curated tractability data that have been integrated from various sources. In addition, computational tools have been developed for predicting target druggability using features derived from gene-level annotations, protein amino-acid sequences and system-level data[5,6]. TargetDB[10] provides a random forest model for tractability prediction using integrated gene-level annotations. DrugMiner[11] employs a neural network model for protein druggability prediction trained on protein sequence composition features. Yu et al.[12] utilised physiochemical properties, protein sequence composition and conservation profiles to train a hybrid deep learning model consisting of a convolutional neural network (CNN) and a recurrent neural network (RNN). A bagging ensemble of support vector machines is developed by Lin et al.[13] using protein sequence composition features while Costa et al.[14] constructed a decision tree-based meta-classifier using topological features derived from system-level gene interaction networks. In addition to the aforementioned generic target druggability approaches, some tools have narrower scopes predicting targets druggability for specific diseases or therapeutic areas such as oncology[15], while other tools are designed for predicting druggability for a specific type of activation or inhibition (e.g. kinease-inhibitor[16]). In addition, there are druggability prediction methods that focus on predicting druggability at binding sites such as TRAPP[17], BiteNet[18], eFindSite[19] and TACTICS[20].

Our work expands on previously published druggability prediction methods by integrating a comprehensive set of druggability-associated features and by providing disease-agonistic and domain-specific models to highlight genes for therapy areas and drug modalities. Specifically, we demonstrate applications of DrugnomeAI in predicting gene druggability for oncology and non-oncology diseases. Our drug modality-specific models predict which genes have properties that make them amenable to modulation by small molecules, monoclonal antibodies and/or proteolysis-targeting chimeras (PROTACs), an emerging drug modality that offers potential improvement over the traditional small-molecule therapeutics and broadens the exploration space for druggable targets.

To the best of our knowledge, this study provides the first ML model for predicting druggability of genes for PROTAC-based therapeutics. PROTACs are bifunctional molecules with two heads: one head binds to the target protein and the other one binds to E3 ubiquitin ligase, a cellular enzyme[21]. The two heads are connected by a linker. PROTACs are protein degraders, i.e. they degrade target proteins instead of inhibiting them, therefore, potentially producing persistent therapeutic effects[22]. Unlike small-molecules that require deep binding sites, PROTACs have a unique ability in targeting proteins with shallow pockets or lacking well-defined binding sites, therefore, targeting proteins that may otherwise be undruggable by traditional approaches[22].

Moreover, researchers can use the DrugnomeAI framework to generate custom and additional disease-specific models by providing user-defined seed genes for training the models. We have validated DrugnomeAI's predictions against successful drug targets and top hits from a phenome-wide association study (PheWAS) of 450 K participants from the UK Biobank (UKB)[23]. Our analysis reveals that protein-protein interaction networks and biological pathway insights are among the most predictive features for binding affinity and therapeutic outcomes. All DrugnomeAI predictions are available at a fully interactive web application to facilitate exploring druggability profiles and visualising key features (http://drugnomeai.public.cgr.astrazeneca.com).

## Results

We sought to infer gene druggability across the whole human exome (19,846 genes) leveraging historical data from known drug targets and other types of evidence around gene tractability, all integrated within the DrugnomeAI ML framework (Fig. 1a). We obtained lists of known or likely druggable genes from the Pharos[24] and Triage[2] resources to train the DrugnomeAI ML models (see Methods). We primarily used two training datasets from Pharos: Tclin (610 genes), consisting of genes that are targets of approved drugs with known mechanism of action, and Tchem (1592 genes), consisting of genes that are targets of compounds included in ChEMBL[25] or DrugCentral[26]. In addition, we used three training datasets from the Triage resource: Tier1 (1411 genes), which comprises

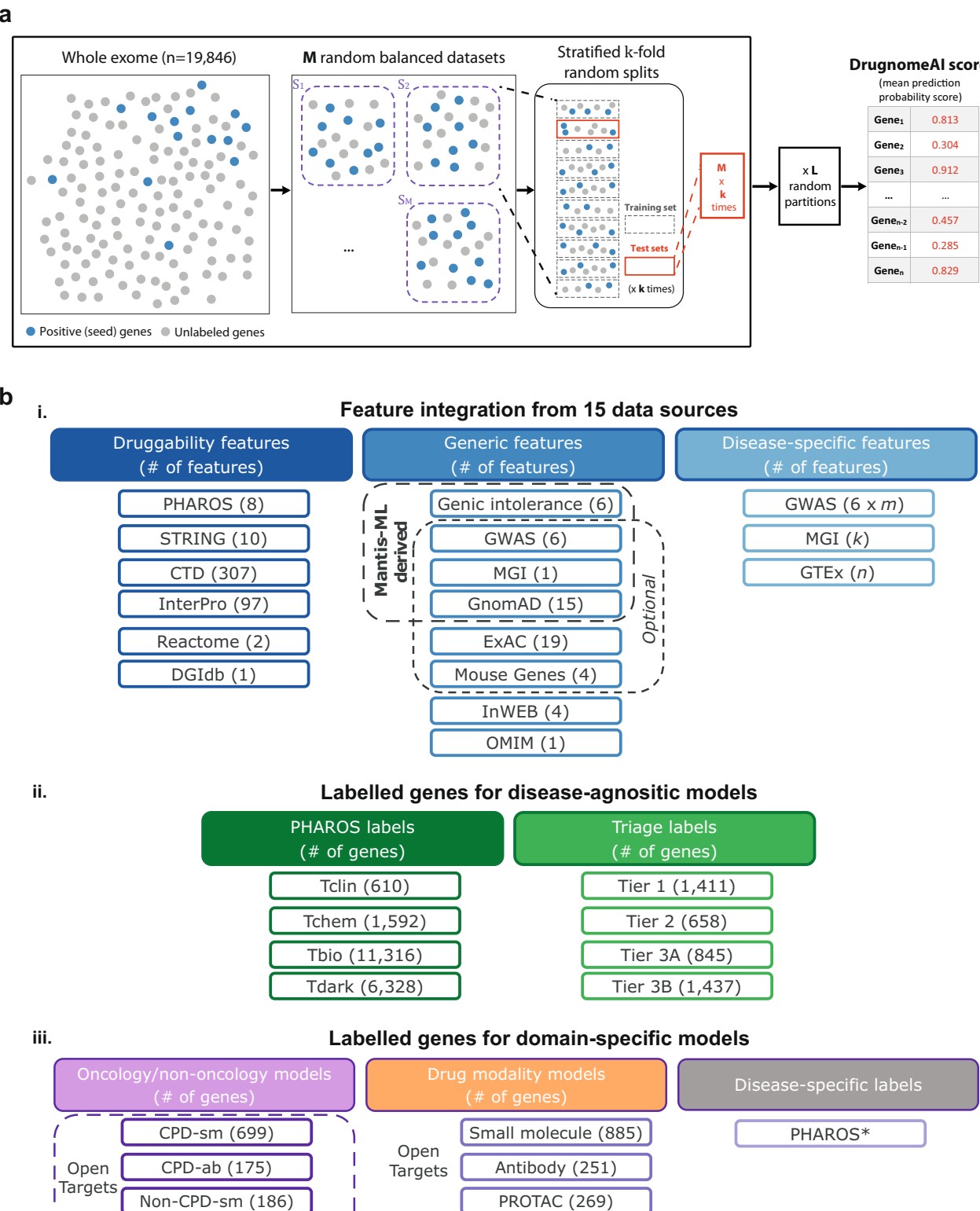

of genes with approved drugs and clinical-phase drug candidates, Tier2 (658 genes), consisting of genes with known bioactive drug-like small molecules and genes with high sequence similarity with approved drug targets, and Tier 3 A (845 genes), which consists of secreted or extracellular proteins that have distant similarity to approved drug targets and gene families not already included in Tier 1 or Tier 2. We trained DrugnomeAI on each of these training sets and extracted druggability predictions based on different types of evidence provided by each labelled dataset.

We tested a range of different druggability and gene-level annotation feature sets during DrugnomeAI training (Fig. 1b).

**Fig. 1 Overview of DrugnomeAI framework and integrated data. a** Illustration of the DrugnomeAI model development workflow. The whole exome (19,846 genes) is split into random balanced subsets of positive (i.e. druggable) and unlabelled (i.e. druggability is unknown) genes. An ensemble of classifiers is generated such that multiple models are trained on each subset with stratified tenfold cross-validation. The process is repeated for L stochastic iterations. The final druggability scores are obtained by averaging the prediction scores from out-of-bag sets across all stochastic iterations from the ensemble models. **b** Data resources integrated in DrugnomeAI. **i** Feature integration from 15 data sources. *m*: number of GWAS-specific terms relevant to a disease; *k*: number of MGI-specific terms relevant to a disease; *n*: number of tissues affected by a given disease. **ii** Data sources of genes druggability labels for disease-agnostic models. **iii** Resources for gene labels used for the domain-specific models (detailed descriptions for each model available in Table 4). *labels are extracted from PHAROS based on the input disease terms.

Specifically, we explored four different feature sets, in increasing order of number of features:

1. "InterPro", comprising of the feature set extracted exclusively from that resource;
2. "Pharos + InterPro" referring to the druggability-specific features only from the respective resources;
3. "All (druggability)" denoting all druggability-specific data sources along with the generic ones inherited from mantis-ml (namely ExAC and Essential Mouse genes data), and
4. "All + Mantis", which in addition to the aforementioned datasets, encompasses other sources utilised in mantis-ml, such as gnomAD, Genic Intolerance, GWAS, and MGI essential data (see Methods).

We evaluated and compared the performance of four classifiers (Random Forest, Extra Trees, Support Vector Machine and Gradient Boosting) across different combinations of labelled datasets and feature sets employed for the predictions. We observe that the Gradient Boosting model consistently outperformed the rest of the classifiers across all configurations of label sets (Supplementary Fig. 2b) and feature sets (Supplementary Fig. 1). Gradient Boosting's hyperparameters were further fine-tuned (see Methods) and it was selected as the default classifier for DrugnomeAI training.

**Analysis of significant druggability-associated features with ablation and Boruta**. In order to select an optimal non-redundant feature set we initially performed a basic ablation analysis. Specifically, we trained DrugnomeAI using three different feature sets, employing more or less extended druggability-associated features, and specifically the: "Pharos + InterPro", "All (druggability)" and "All + Mantis" feature sets (already described in the previous section). AUC scores achieved by the "Pharos + InterPro" dataset were either identical or comparable with those extracted by the more extended "All (druggability)" and "All + Mantis" feature sets (Supplementary Fig. 1). Thus, we selected the "Pharos + InterPro" as the default feature set for DrugnomeAI to eliminate any non-informative redundancy from the more extended feature sets. Next, we performed feature importance analysis with Boruta algorithm[27] for the Tclin (Fig. 2c) and Tier 1 labelled datasets (Supplementary Fig. 2a). Boruta is an iterative feature selection method to determine if a feature has a statistically robust predictive power. It compares the predictive power of each feature against randomised versions of the original feature set (called "shadow" features), using a Random Forest as the base model for classification. Weak features (i.e. features proved statistically less relevant than the maximum of "shadow" features) are removed. Once the model converges, a "confirmed" set of features (i.e. features that are considered predictive) are identified, and are ranked based on Z-scores representing importance scores (see Methods).

For both models, the most important features were related to protein-protein interactions based on the DGIdb[28], InWeb[29], Reactome[30] and STRING[31] networks. This is consistent with existing literature that has demonstrated that interaction partners

of druggable genes are also more likely to be druggable[2]. In addition, protein-protein interactions are linked to biological and pathological processes and, recently, protein-protein interactions have gained increasing attention as drug targets due to their potential for selectively modulating specific pathways[32,33]. Upon performing principal component analysis (PCA; Supplementary Fig. 3) of the Tclin and Tier 1 datasets, we observed that the first principal components capture only ~4.5% of the variance, indicating the presence of highly non-linear relationships between the features (Supplementary Fig. 10).

After selecting the optimal feature set, we investigated its performance across different classifiers for an array of labelling variants. We provide detailed AUC score breakdown across the various configurations (Supplementary Fig. 2b). The Gradient Boosting classifier consistently and significantly outperformed the other algorithms across all the examined configurations. Specifically, we applied the DeLong test to compare the AUC scores attained by the Gradient Boosting against the respective performance from all other classifiers (Random Forest, Extra Trees, Support Vector Classifier and Deep Neural Net) based on the Tclin and Tier1 labelled datasets (Supplementary Fig. 24). We observe that Gradient Boosting significantly outperforms all other classifiers for both the Tclin and Tier1 labelled datasets (Tclin dataset – DeLong test $p$ values of Gradient Boosting vs Random Forest: $p = 4.34 \times 10^{-18}$, Extra Trees: $p = 1.01 \times 10^{-18}$, SVC: $p = 2.44 \times 10^{-10}$, DNN: $p = 6.58 \times 10^{-12}$; Tier1 dataset – DeLong test $p$ values of Gradient Boosting vs Random Forest: $p = 5.04 \times 10^{-29}$, Extra Trees: $p = 2.83 \times 10^{-30}$, SVC: $p = 3.32 \times 10^{-15}$, DNN: $p = 1.71 \times 10^{-10}$). Finally, Gradient Boosting's AUC score is characterised by the lowest variance which means that a choice of a labelled set does not influence noticeably the classifier performance (Fig. 2a). As for the labelled dataset variants, the highest results were obtained using Tclin and Tier 1 (AUC = 0.99 and 0.97, respectively; Fig. 2b).

**Validation and exploration of DrugnomeAI top hits**. Since the best performance was achieved using a Gradient Boosting model trained with the Tclin or Tier 1 label sets, we use these predictions as our reference models for further analyses (referenced as DrugnomeAI-Tclin and DrugnomeAI-Tier1, respectively). We obtained the top 5% of genes ranked by DrugnomeAI-Tclin and DrugnomeAI-Tier1, each consisting of 992 genes (Supplementary Data 1). Notably, there is 63% (621 genes) overlap between the two sets (Supplementary Data 2).

**Top DrugnomeAI hits with clinical evidence**. We conducted a systematic review across all clinical development activities to identify genes that have been implicated as targets in therapeutic drug development (i.e. genes that have been selected for clinical development; see Methods) among the top 5% of genes ranked by the DrugnomeAI-Tclin and/or DrugnomeAI-Tier1. We grouped these genes into 20 rank intervals, each containing ~992 genes. We found that genes ranked in the top 5% by DrugnomeAI-Tclin were significantly enriched among genes selected for clinical

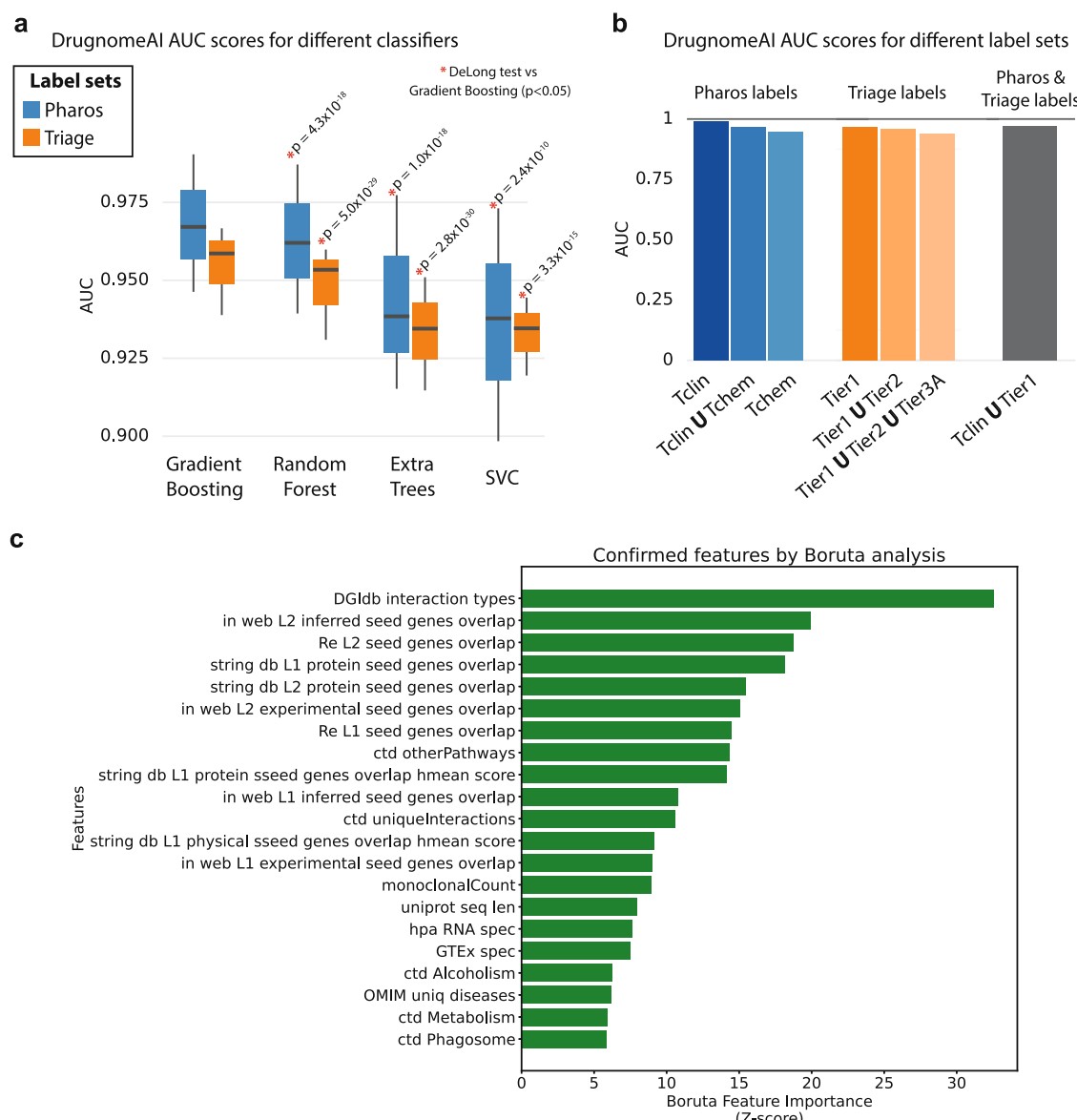

**Fig. 2 Analysis of DrugnomeAI models' predictive performance and top contributing features. a** DrugnomeAI AUC score distribution across different classifiers and labelling variants utilising the druggability-specific dataset (the statistical significance of Gradient Boosting outperforming the other classifiers has been calculated using DeLong test, with the corresponding *p* values provided above each barplot). **b** AUC scores (with Gradient boosting) across different labelling variants utilising the druggability-specific dataset. **c** List of druggability-associated features extracted by the Boruta feature selection algorithm (as "Confirmed" features) on the Tclin dataset.

development (Odds Ratio = 132.78, Fisher's exact test $p < 1 \times 10^{-308}$; Fig. 3b, Supplementary Data 3, Supplementary Figs. 4, 5). 753 genes (63% of the interval) ranked in the top 5% by DrugnomeAI-Tclin and 268 genes in the 5–10% rank interval are supported by prior clinical development efforts (Fig. 3a). We observe similar levels of strong enrichment among genes ranked by DrugnomeAI-Tier1 (Fig. 3c, d). Remarkably, based on the cumulative distribution function we observe that 25% of top ranked genes by DrugnomeAI explain 95% of genes supported by clinical evidence (Fig. 3e), and 80% of genes supported by clinical evidence are ranked among the top 10% genes by DrugnomeAI.

We conducted further analysis of the top 5% genes ranked by DrugnomeAI-Tclin and DrugnomeAI-Tier1. 76% and 61% of genes in Tclin and Tier1-based predictions, respectively, have been selected for clinical development. Furthermore, 627 (63%) and 475 (48%) genes from the Tclin and Tier1-based predictions, respectively, are targeted by small molecules. Of these genes, we

found clinical trials had progressed into phase IV for 501 (51%) and 346 (35%) genes in Tclin and Tier1-based, respectively (Fig. 4a, b). We also analysed the therapeutic areas that the top 5% genes ranked by DrugnomeAI models have been implicated with, and observed that the majority of those genes have been selected for clinical development for genetic diseases, cell proliferation disorders (CPD), nervous system diseases and immune system diseases targeted by small molecules or monoclonal antibodies (Supplementary Fig. 11).

**Genes with no prior evidence in clinical development**. In the previous section, we demonstrated that there are 239 and 387 targets among the top 5% predicted hits from the Tclin and Tier1-based DrugnomeAI models, respectively, that do not have any clinical trials data associated with them (Fig. 4a, b). These genes are predicted by DrugnomeAI to be druggable but do not yet have drugs in clinical development. We identified potential

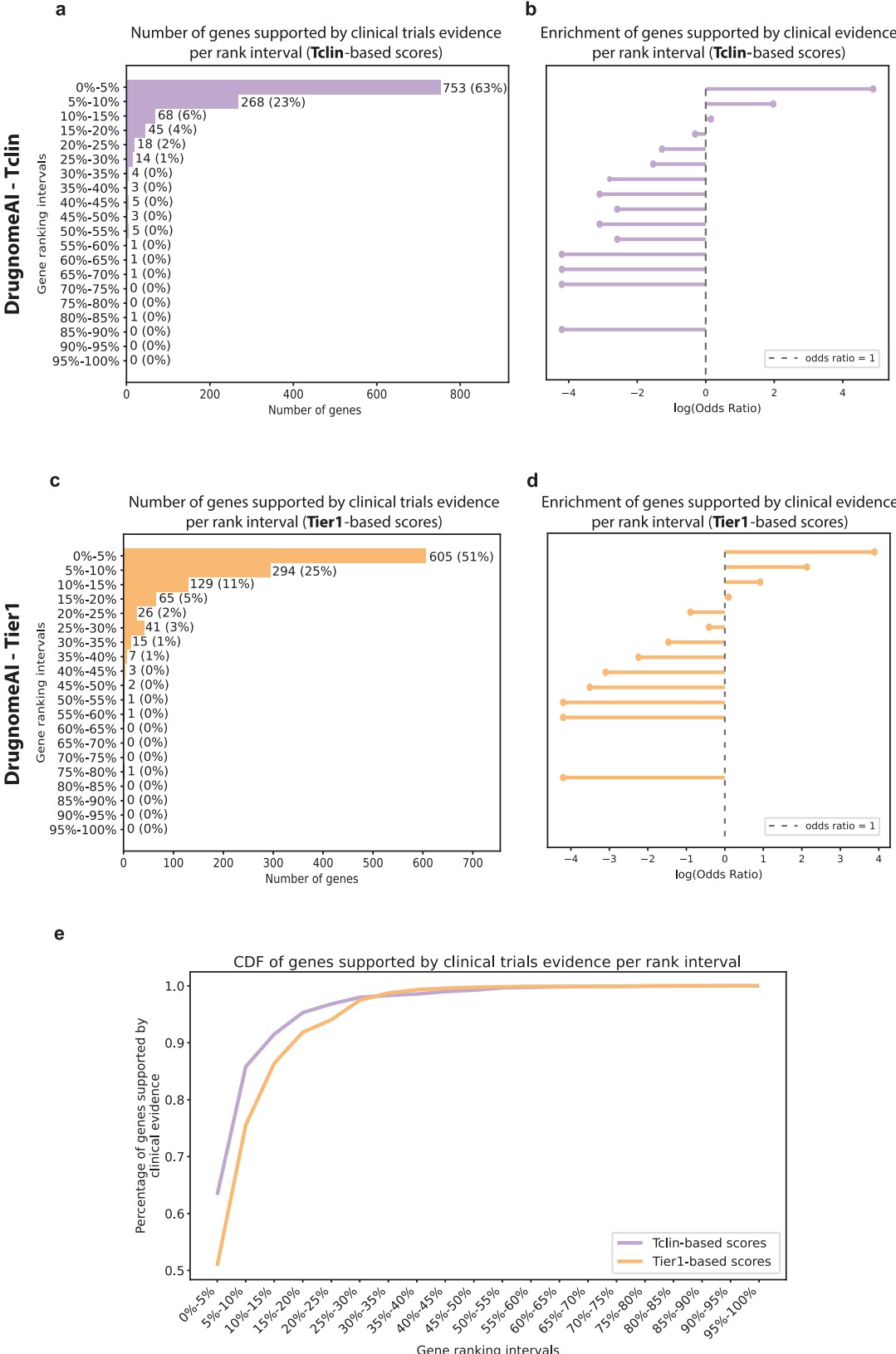

**Fig. 3 Validation of DrugnomeAI ranked genes using clinical evidence.** Number of genes ($n = 19{,}846$) supported by clinical evidence per rank intervals based on predictions of (**a**) DrugnomeAI-Tclin and (**c**) DrugnomeAI-Tier1. 0–5% consists of genes ranked in the top 5% whereas 95–100% contains genes ranked in the bottom 5%. Enrichment of genes supported by clinical evidence in each rank interval based on predictions of (**b**) DrugnomeAI-Tclin and (**d**) DrugnomeAI-Tier1. Larger odds ratio values indicate higher enrichment. **e** Cumulative distribution function (CDF) plot of genes supported by clinical evidence per rank interval.

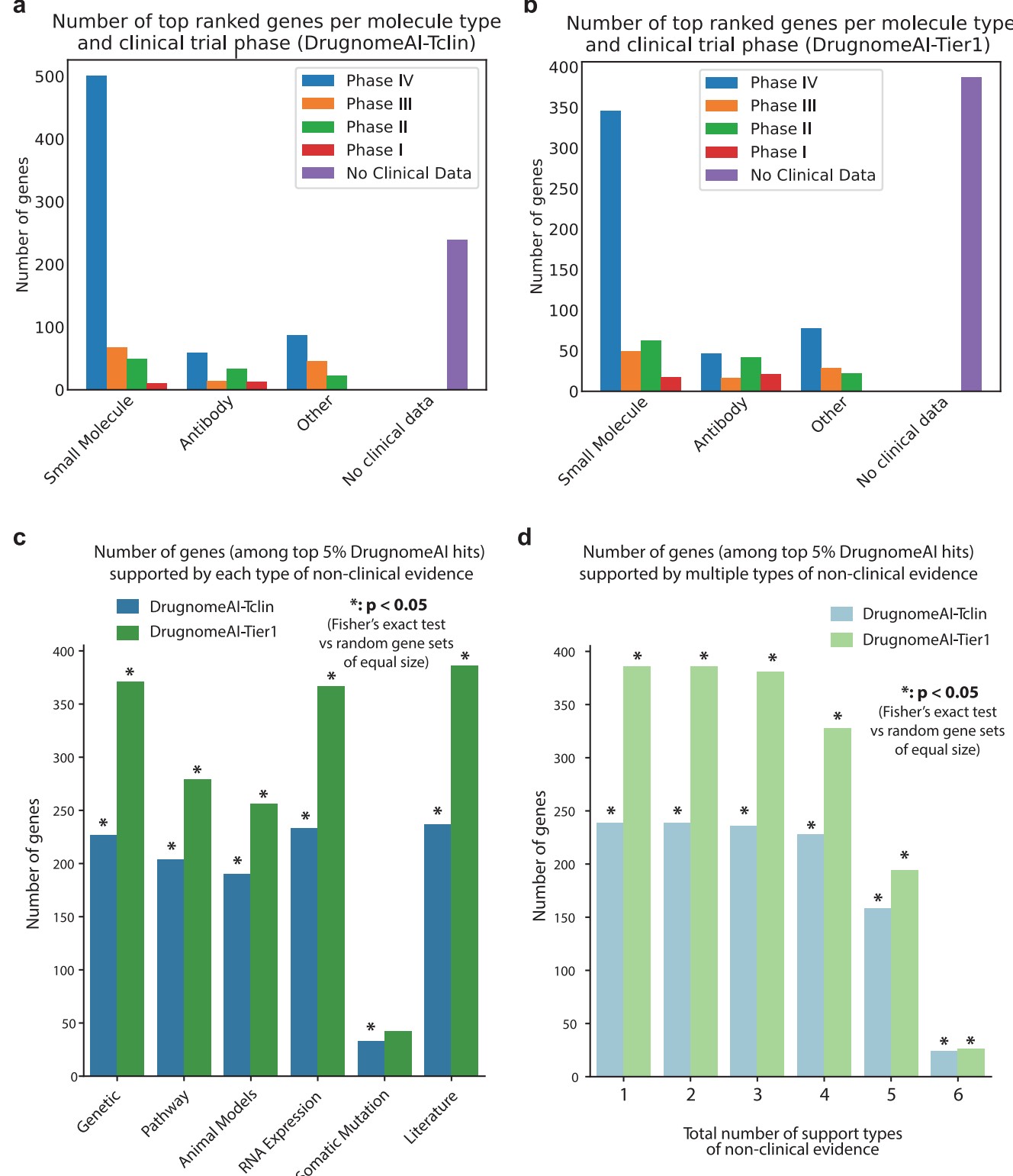

**Fig. 4 Clinical and non-clinical evidence for the top 5% of genes ranked by DrugnomeAI.** Clinical evidence available for the top 5% genes ($n = 992$) ranked by (**a**) DrugnomeAI-Tclin and (**b**) DrugnomeAI-Tier1. Each bar indicates the number of genes targeted by each molecule type per clinical trial phase. **c** Number of genes among the top 5% DrugnomeAI Tclin-based and Tier1-based predictions satisfying each distinct type of non-clinical evidence. **d** Number of genes among the top 5% DrugnomeAI Tclin-based and Tier1-based predictions satisfying multiple types ($x = 1,2,..6$) of non-clinical evidence. Asterisks (*) denote that the respective gene sets are significantly enriched for each type or set of types of non-clinical evidence compared to 10 random gene sets of equal size (the median $p$ value is eventually used to assess significance in each case).

associations between these genes and diseases using non-clinical evidence (i.e. associations between genes and diseases that are not supported by clinical trials). We used six types of non-clinical evidence: genetic, animal models, somatic mutations, RNA expression, pathways, and literature (Fig. 4c; Supplementary Fig. 28. see Methods). We performed enrichment analyses for the top-ranking genes from Tclin and Tier1 without clinical support, against one or more types of support accompanying each of them, performed via Fisher's exact test against multiple random subsets of genes (null subsets). For each enrichment analysis, we report the median $p$ value achieved across 10 iterations against random (null) genes sets (Fig. 4d, Supplementary Fig. 27). We found all 239 (Tclin-based predictions) and 386 out of 387 (Tier1-based predictions) genes to be associated with diseases and supported by at least two types of non-clinical evidence (Tclin - Fisher's exact $p = 1.5 \times 10{-08}$; Tier1 - Fisher's exact $p = 5.2 \times 10{-10}$; Fig. 4d, Supplementary Fig. 27). Large proportions of the Tier1 and Tclin top ranking genes without clinical evidence are further supported by three, four or even five types of support, and significantly so compared to random gene sets (Fig. 4d, Supplementary Fig. 27). Finally, there are 24 genes from Tclin-based predictions (Fisher's exact $p = 2.4 \times 10{-3}$) and 26 genes from Tier1-based predictions (Fisher's exact $p = 1.3 \times 10{-2}$) that are supported by six types of evidence (Fig. 4d, Supplementary Figs. 9, 27). While all levels of support are statistically significant, we observe that for genes supported by six types of evidence, Fisher's exact test $p$ value are relatively lower compared to the other analyses. This is expected though due to the smaller number of genes supported by all six types of non-clinical evidence. Overall, it's notable that the top hits predicted by DrugnomeAI (without having prior clinical evidence) are highly and significantly enriched for multiple types of non-clinical evidence, suggesting that they are more likely to be biologically relevant with regards to their druggability potential.

We then expanded the enrichment analysis for non-clinical evidence across all genes ranked by the DrugnomeAI-Tclin and DrugnomeAI-Tier1 models. Overall, top ranked genes by the two models are significantly enriched among genes supported by genetic evidence (DrugnomeAI-Tier1: Odds Ratio = 5.8, Fisher's test $p$ value = $9.35 \times 10^{-38}$ and DrugnomeAI-Tclin: Odds Ratio = 4.6, Fisher's test $p$ value = $4.64 \times 10^{-32}$). In addition, there is high enrichment among genes supported by the other five types of non-clinical evidence (Supplementary Data 4, Supplementary Figs. 6, 7).

Next, we explored the features of genes not previously pursued clinically to examine whether there are any identifiable traits that would distinguish them from genes selected for clinical development. We plot the kernel density estimate of the top 20 features from Tclin and Tier1 and employ the Chi-squared statistical test to compare the distribution of any of these features between genes with or without clinical evidence (Supplementary Figs. 25, 26). Top ranked features include monoclonal count, antibody count, protein sequence length, and DGIdb interaction types ($p$ value < $1 \times 10^{-308}$) where we observe that genes without clinical evidence have on average smaller values than genes that have been selected for clinical development. Notably, the CTD processes "decreases metabolic processing" and "increases uptake" have non-zero distributions among the genes without clinical evidence and are significantly different from the distributions of genes with clinical support. This may suggest that genes with no prior clinical evidence may be involved in metabolic pathways which are highly complex or more challenging to target. Other significant features include associated pathways and interactions from the Comparative Toxicogenomics Database (CTD). For example, we observe that "CTD increased cleavage" is highly present among genes that have been

selected for clinical development but is depleted in genes without clinical evidence. That is expected as cleavage is one of the most established steps involved in drug mechanism of action, such as antibody drug conjugates for treating treating tumours[34], and seems to have already been studied and covered extensively among known drug targets (detailed explanation of all these features is available in Supplementary Data 5).

**Enrichment with significant gene hits from large-scale PheWAS studies**. We investigated the overlap between the top 5% DrugnomeAI predictions and the highly ranked genes from large-scale phenome-wide association studies (PheWAS) for binary and quantitative traits extracted from 450 K samples from the UKB cohort[23] (see Methods). We analysed the enrichment of top 5% genes ranked by DrugnomeAI models and supported by clinical evidence with genes achieving genome-wide significance ($p$ value < $5 \times 10^{-8}$) from PheWAS in UKB (Fig. 5, Supplementary Fig. 8). We observe significant enrichment of top 5% genes ranked by DrugnomeAI-Tclin among the top PheWAS for binary traits (Odds Ratio = 2.9, Fisher's exact test $p$ value = $1.69 \times 10^{-5}$) and for quantitative traits (Odds Ratio = 2.5, Fisher's exact test $p$ value = $1.56 \times 10^{-7}$). Similarly, there is a significant enrichment of highly ranked DrugnomeAI-Tier1 predictions among top genes from PheWAS binary traits (Odds Ratio = 3.0, Fisher's exact test $p$ value = $4.63 \times 10^{-5}$) and PheWAS qualitative traits (Odds Ratio = 3.0, Fisher's exact test $p$ value = $9.53 \times 10^{-10}$).

**Enrichment of top DrugnomeAI genes with OMIM disease annotations**. We then assessed how genes associated with OMIM diseases are ranked by DrugnomeAI models (Supplementary Fig. 23, Supplementary Data 6). We observe that genes associated with OMIM diseases are also significantly enriched among the top 5% ranked genes by DrugnomeAI-Tclin (Fisher's exact test $p$ value = $6.05 \times 10^{-110}$, Odds Ratio = 4.6) and DrugnomeAI-Tier1 (Fisher's exact test $p$ value = $6.55 \times 10^{-77}$, Odds Ratio = 3.6). Specifically, 506 (51%) and 452 (45%) genes ranked among the top 5% DrugnomeAI-Tclin and DrugnomeAI-Tier1 hits, respectively, have been associated with OMIM diseases. That suggests that a relatively large proportion of genes predicted to be highly druggable may also have high likelihood to be biologically relevant and carry out a disease-specific therapeutic effect.

**Benchmarking against other druggability prediction methods**. We sought to explore how DrugnomeAI compares with published methods for druggability prediction, focusing on methods that can perform disease-agnostic exome-wide druggability predictions. We selected three tools for this task, which provide either pre-calculated prediction scores or a code repository for reproducing their models: (1) TargetDB, a recently published tool employing a random forest model for tractability prediction[10], (2) a recently published deep learning model by Yu et al.[12] for protein druggability prediction, and (3) a decision tree-based meta classifier by Costa et al.[14] for genome-wide prediction of morbid and druggable genes.

To assess the enrichment for top predictions from each model, we employed two data sources for validation as independent reference sets: the Open Targets tractability data for small molecules and antibodies and a list of genes with approved drugs from King et al.[35]. We investigated whether the top 5% genes (top 992 genes per model) from DrugnomeAI, TargetDB and the models by Yu et al.[12] and Costa et al.[14] preferentially overlap with each of the validation datasets. Remarkably, we observed that DrugnomeAI-Tclin has the highest overlap with the validation datasets. The top-ranked genes from DrugnomeAI-Tclin overlap

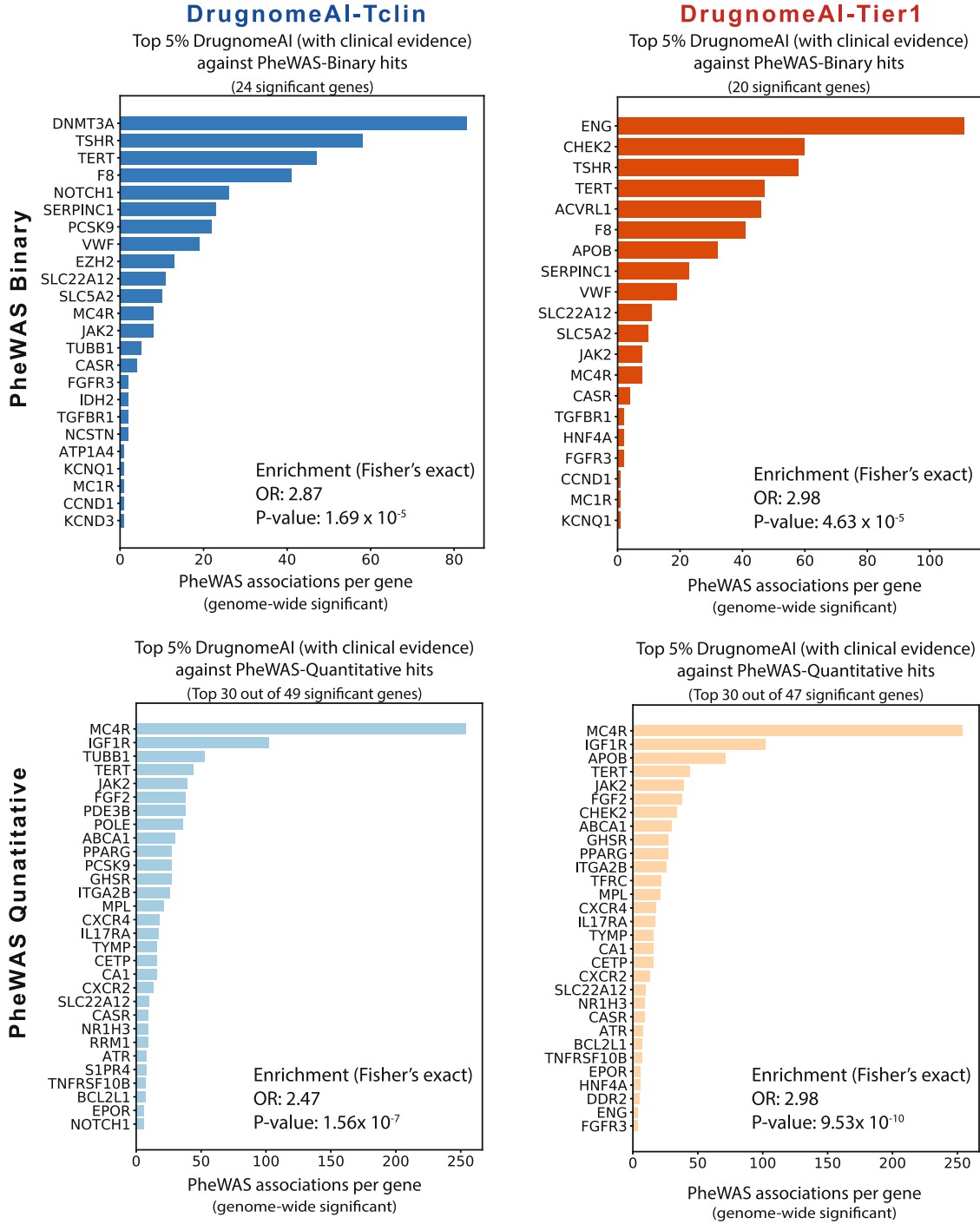

**Fig. 5 Enrichment of top 5% genes ($n = 992$) ranked by DrugnomeAI-Tclin and DrugnomeAI-Tier1 and supported by clinical evidence among the top UKB PheWAS hits for binary and quantitative traits.** Genome-wide significant hits ($p < 5 \times 10^{-8}$) have been considered from the PheWAS analysis on 450 K samples from UKB. While the common top hits from DrugnomeAI and PheWAS are sorted by the number of significant hits in PheWAS for visualisation purposes, it is expected that many of the associated phenotypes may be highly correlated.

with the validation datasets by 35%, 29%, and 149% more than the top-ranked hits from TargetDB, Costa et al. and Yu et al., respectively (Fig. 6). We also observe that the DrugnomeAI-Tclin overlap with approved drug targets from King et al. is statistically significant compared to the overlap from TargetDB (Fisher's exact test $p$ value $= 3.9 \times 10^{-15}$, Odds Ratio $= 2.3$), Yu et al. (Fisher's exact test $p$ value $= 3.6 \times 10^{-79}$, Odds Ratio $= 17.2$), and Costa et al. (Fisher's exact test $p$ value $= 2.3 \times 10^{-10}$, Odds Ratio $= 1.9$) with the same validation dataset (Supplementary Data 7).

We also performed a stepwise hypergeometric test to assess the enrichment of top predictions by DrugnomeAI, TargetDB, the Costa et al.[14] and Yu et al.[12] models among the validation datasets (Supplementary Fig. 12). To further quantify the enrichment, we calculated the area under the hypergeometric curve (AUC) of the enriched region ($p$ value $< 0.05$) (Supplementary Data 8). In three out of the seven test cases (Supplementary Fig. 12), DrugnomeAI-Tclin demonstrated higher enrichment than DrugnomeAI-Tier1, TargetDB, Costa et al., or Yu et al. For genes targeted by small molecules, top

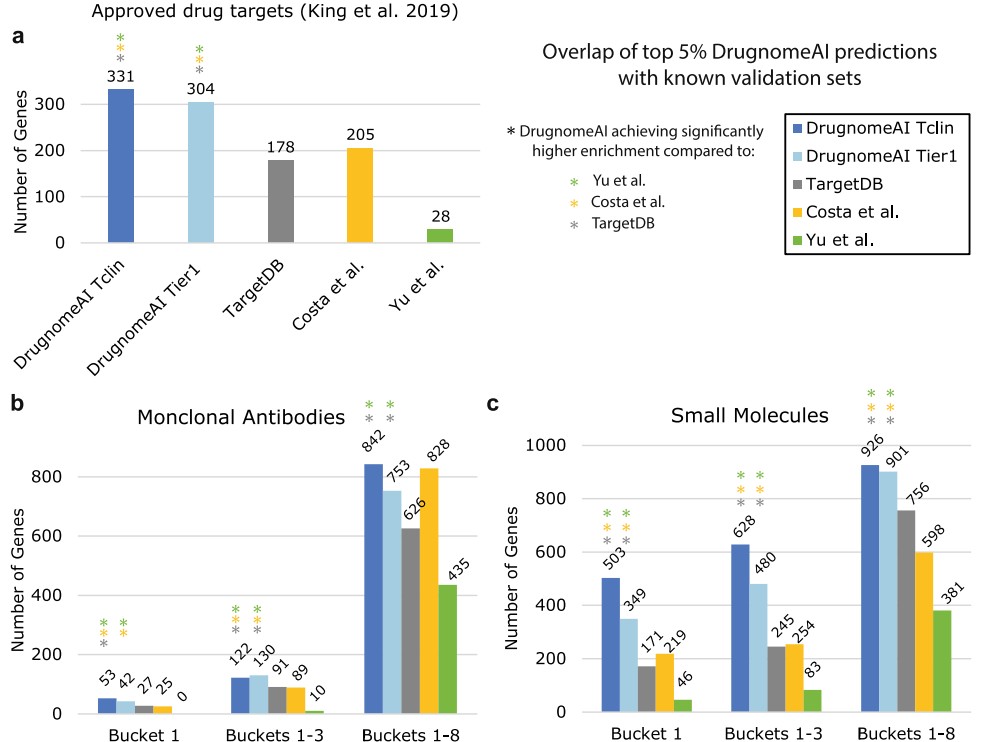

**Fig. 6 Overlap between top 5% genes (n = 992) ranked by DrugnomeAI-Tclin, DrugnomeAI Tier1, TargetDB, Costa et al. and Yu et al. across three validation datasets.** Validation of the five models across three validation datasets: (**a**) Approved drug targets (King et al., 2019 dataset). **b** Open Targets druggability dataset for monoclonal antibodies. **c** Open Targets druggability data for small molecules. DrugnomeAI has a significantly more enriched overlap in the majority of pairwise comparisons (39 out of 42 comparisons). Coloured (with grey, yellow and green) asterisks indicate where DrugnomeAI models achieves significantly higher enrichment with each known validation set compared to TargetDB, Costa et al., and Yu et al., respectively (see also Supplementary Data 7).

predictions by DrugnomeAI-Tclin were significantly enriched for genes with approved drugs in Bucket 1 from Open Targets (AUC was 23-fold higher than TargetDB and Costa et al.) and genes selected for clinical development in Buckets 1–3 (AUC was 10-fold and 13-fold higher than TargetDB and Costa et al., respectively). In addition, we observed significant enrichment among genes with approved drugs in King et al. dataset (AUC was 3-fold and 2-fold higher than TargetDB and Costa et al., respectively). For genes targeted by monoclonal antibodies, DrugnomeAI-Tier1 top predictions are significantly enriched for genes in Buckets 1–8 (area under curve was 372-fold higher than TargetDB). Top predictions by TargetDB are more enriched among genes in Buckets 1–8 targeted by small molecules (AUC is 3-fold higher than DrugnomeAI-Tclin and Costa et al.). DrugnomeAI models exhibited lower enrichments for genes with approved monoclonal antibodies in Bucket 1 and genes selected for clinical development in Buckets 1–3. This could be due to the small number of genes in these datasets that achieved significant enrichment. In addition, the training sets (Tclin and Tier1) are likely skewed towards genes targeted by small molecules. Finally, the top hits by the Yu et al. model have low enrichment with zero AUC scores in all cases.

**Therapeutic modality-specific models**. Apart from the generic DrugnomeAI models, we developed models specific to three drug modalities: small molecule, monoclonal antibody, and PROTAC, which are trained on genes known to already be amenable by each modality type, respectively. We selected these modalities since small molecule and monoclonal antibody inhibitors are two of the main types of targeted therapies, and PROTAC technology is an emerging modality that can overcome some of the drawbacks of

small molecule-based therapies[36]. In addition, these molecules tend to successfully target different types of proteins. For example, small molecules are quite amenable to targeting intracellular proteins while monoclonal antibodies can primarily target extracellular proteins[37]. Therefore, obtaining granular druggability scores for each therapeutic modality could help prioritise targets that are likely to be druggable by a specific drug modality. However, our framework is generic in nature, and it can be extended to other therapeutic modalities once a sufficient volume of appropriate training data is available.

We tested four classifiers (gradient boosting, random forest, SVC, and extra trees) per drug modality. Although the four classifiers achieved comparable performance in target predictability (AUC ≥ 0.94), gradient boosting models outperformed all other classifiers achieving AUC scores of 0.98, 0.99, and 0.97 for antibody, small molecule, and PROTAC modalities, respectively (Supplementary Fig. 13). We also observed high correlation of gene probability predictions across the four classifiers reaching Pearson's r scores of up to 0.93, 0.94, and 0.95 for small molecules, monoclonal antibodies, and PROTACs modalities, respectively (Supplementary Fig. 14).

Exploring the top 50 genes ranked per drug modality reveals several novel genes (i.e. unlabelled genes with high rankings and not among the seed genes in the model training). There were 17 and 16 novel genes among the top 50 genes ranked for antibody and PROTAC modalities, respectively, while all the top 50 genes by the small molecule model were known genes (Supplementary Fig. 16). We also assessed whether the antibody-specific DrugnomeAI predictions were preferentially under-represented for intracellular proteins, which are difficult or impossible to be accessed by this modality type. Specifically, we found that only

182 out of the 1181 positive DrugnomeAI predictions from the antibody-specific model (probability score > 0.5) are found exclusively in the intracellular space, which is significantly lower than the overall representation of intracellular proteins in the rest of the exome (Fisher's exact test $p = 3.1 \times 10^{-139}$, Odds Ratio = 0.17), based on another 7544 intracellular proteins found in the remaining 14,601 proteins of the exome with known information about their cellular localisation (as derived from Open Targets[8]). For reference, the training set for the antibody-specific DrugnomeAI model (as derived from Open Targets), showed a similar under-representation of intracellular proteins (22 out of the total 230) with information about their cellular localisation (Fisher's exact test $p = 3.4 \times 10^{-38}$, Odds Ratio = 0.11).

Schneider et al.[38] describes a set of 1067 genes as potential PROTAC targets, not previously described in literature, that are also distinct from the genes we used for training our PROTAC-based DrugnomeAI model. We explored how these genes are ranked by the DrugnomeAI PROTAC model (Supplementary Fig. 17) and, notably, observed high enrichment with 287 (27%) of these genes being ranked in the top 5% (Fisher's exact test $p$ value $= 6.7 \times 10^{-138}$, Odds Ratio = 9.5).

**Oncology and non-oncology specific DrugnomeAI models**. Considering that targets for oncology diseases have different regularity requirements for safety and efficacy, we examined whether genes that have been selected for development in the oncology space have distinct properties from genes targeted for other disease areas. To this end, we explored genes previously selected for development for CPD, which include cancerous and pre-cancerous conditions as well as neoplastic diseases and hyperplasia, as well as a narrower set consisting of only cancer-related genes. Therefore, we investigated five scenarios: (1) "CPD-sm" and (2) "CPD-ab", which are trained on CPD genes targeted by small molecules and antibodies, respectively; (3) "non-CPD-sm" and (4) "non-CPD-ab", which are trained using genes targeted by small molecules and antibodies, respectively, and these genes have not been selected for development for CPDs; and (5) "cancer-sm" model using genes in cancer cell lines linked with small molecules.

In all five scenarios, the classifiers that were tested achieved high performance (AUC > 0.93) (Supplementary Fig. 18). Gradient boosting again outperformed all other classifiers achieving AUC scores of 0.99, 0.98, 0.98, 0.98, and 0.96 for cases (1)–(5), respectively. Overall, there is high correlation of gene probability predictions across the four classifiers reaching Pearson's r scores of up to 0.95 for the "cancer-sm" case and 0.93 for the remaining cases (Supplementary Fig. 19), selecting again Gradient Boosting as the default classifier. We then aimed to determine whether there are any novel genes among the top 50 genes ranked by DrugnomeAI in each case (Supplementary Fig. 20), identifying 22 and 30 novel genes for the CPD-ab and non-CPD-ab models, respectively.

**Significant features analysis of domain-specific DrugnomeAI models**. We sought to explore the most important features for predicting druggable genes for each modality. Analysis of confirmed features by the Boruta algorithm shows that features derived from protein-protein interaction networks ("seed genes overlap hmean score", "inferred seed genes overlap", "experimental seed genes overlap", "Re seed genes overlap", and "StringDB protein genes overlap") are high contributors for the three modalities (Supplementary Fig. 15). These features represent the ratio of known druggable genes interacting with a candidate target from InWeb and StringDB (see Methods). In addition, features derived from interaction data, such as DGIdb

interaction types (number of gene-drug interactions from DGIdb) and CTD unique interactions (number of unique chemical-gene interactions from CTD) as well as monoclonal count (number of monoclonal antibodies for a target) are also high contributors for all three drug modalities. These features indicate that the druggability problem can better be addressed from a systems biology point of view instead of pursuing each target individually. Moreover, associated pathways from CTD is a top feature for small molecule and antibody modalities, while protein-coding sequence length is a top feature for the small molecule and PROTAC modalities (detailed explanation of each feature is available in Supplementary Data 5).

Similar to the modality-specific models, Boruta analysis showed that features from protein-protein interaction networks, associated pathways, unique interactions, monoclonal count, and sequence length were among the top features for the oncology and non-oncology specific DrugnomeAI models (Supplementary Fig. 21). In addition, we observed features associated with pathways from CTD (representing the presence or absence of a gene in a given pathway) are top contributors for druggability prediction. Specifically, CTD apoptosis is a high feature for "CPD-sm" and CTD Phagosome is among the top features for "CPD-ab" while CTD Metabolism is a top feature for both of the oncology-related small molecule modalities ("CPD-sm" and "cancer-sm").

## Discussion
Target selection is a crucial step in the drug discovery pipeline and selecting the right targets early on in development has a huge impact on the success rate of late-stage clinical trials. Here, we introduce DrugnomeAI to support target selection process by quantifying a probability score per gene that represents its likelihood as a good candidate target for small molecule, monoclonal antibody, and PROTAC development. We demonstrate DrugnomeAI's high predictive power even when trained on a small set of positive labelled genes. We have illustrated the tool's broad application scope using the disease-agnostic models in addition to therapeutic area (oncology and non-oncology) and therapeutic modality (small molecule, monoclonal antibody, and PROTAC) stratified models. In addition, we provide a web-based resource to facilitate exploring the generated druggability profiles and corresponding key properties of druggable genes.

Our work demonstrates that gene properties at the systems biology level derived from protein-protein interaction networks are among the top contributors in predicting druggability. While we have included features around kinase domains and ligand binding from InterPro and CTDbase, respectively, these are not selected among the top features. This may be due to the small labelled dataset available to highlight this type of contribution. Therefore, in this study, we capture the macroscopic signals at a systems biology level around predicting targets druggability rather than identifying the refined fundamental principles that drive druggability.

Opportunities for future expansion involve integrating graph CNNs[39] or other graph embeddings approaches[40] to capture higher resolution information from protein-protein interaction networks. Another possible expansion of our work involves targeting pocket-level predictions by investigating druggability of binding sites for small molecules, epitopes for antibody-based therapies, or regulatory elements such as transcription factors. This requires incorporating features specific to therapeutic modalities and binding pockets.

Since the training sets mainly consist of genes that have approved drugs or drugs in clinical trials, a major point of concern was the possibility of overfitting the models to the

characteristics of these genes. However, our analysis shows that there are highly ranked genes that have not been selected for drug development programs yet have strong associations with diseases. These genes may have not been selected for drug developmental programs so far either because they are associated with less common mechanisms of disease or require different therapeutic modulation approach.

A limitation of DrugnomeAI and of other data-driven approaches is the tendency to overlook under-studied genes since the feature set may not include rich annotations about those genes. This leads to models prioritising genes that are similar to previously known and well-studied drug targets and not necessarily identify novel targets that act on different mechanisms. While the goal of the study is to develop models that capture existing knowledge of known drug targets, a notable point of expansion for this work is identifying novel targets among under-studied genes. This could be achieved by learning from sequence data directly instead of an annotation-driven approach.

Druggability is not an innate property of a gene and a target's druggability can be disease- and drug-dependent. Druggability of a gene is a complex property that is influenced by the pharmacodynamics and pharmacokinetics in vivo along with safety, commercial and regulatory considerations. Another important factor affecting druggability is the direction of modulation, i.e. whether a target must be inhibited or activated to change its biological function and corresponding phenotypic effects. Therefore, comprehensive profiling of targets' druggability, ligandability, inhibitability and activatability will further help us expand our knowledge around the druggable genome and enable us to discover novel targets.

## Methods
**Prediction model architecture**. DrugnomeAI adopts the mantis-ml framework, implementing stochastic semi-supervised learning by splitting the dataset into balanced datasets consisting of positive and unlabelled genes (Fig. 1a). In each stochastic iteration, an ensemble of models is trained on each of the balanced datasets with 10-fold cross-validation. The final prediction score per gene is the average score derived from all models and across all iterations (whenever each gene was part of an out-of-bag test set). Detailed description of the architecture is available in Vitsios et al.[7] Similar to mantis-ml, DrugnomeAI consists of several modules for data pre-processing, semi-supervised and unsupervised learning, feature analysis, and post processing[7]. DrugnomeAI extends on mantis-ml by incorporating druggability specific-features (explained in the next subsection) and support of an additional classifier (Naïve Bayes). Generic and disease-specific resources used in DrugnomeAI have been derived from the original mantis-ml framework. The main parameters of the framework (such as number of stochastic iterations and balancing ratio of positive vs unlabelled data) have also been inherited from mantis-ml, and are available at the DrugnomeAI repo: https://github.com/astrazeneca-cgr-publications/DrugnomeAI-release/blob/master/drugnome_ai/conf/.config. Gradient Boosting's parameters have been specifically optimised for the DrugnomeAI runs (see also "Sensitivity analysis of Gradient Boosting hyperparameters" section in Methods). The parameters used in developing Random Forest, Extra Trees, SVC and DNN models are inherited from mantis-ml, and are available here: https://github.com/astrazeneca-cgr-publications/DrugnomeAI-release/blob/master/drugnome_ai/modules/supervised_learn/classifiers/ensemble_lib.py.

### Druggability-specific resources
*Pharos*. Pharos[24] is a web interface to browse the Target Central Resource Database (TCRD, http://juniper.health.unm.edu/tcrd) and is publicly available at https://pharos.nih.gov. We integrate data from Pharos referencing information directly from TCRD which constitutes an information source for the Druggable Genome amassing data from a variety of resources on human drug targets (including the Harmonizome[41], Jensen Lab datasets, EBI data sets, the Drug Target Ontology). For the purpose of DrugnomeAI we incorporate data on target antibodies, protein-protein interactions, tissue specificity, sequence size retrieved from UniProt[42], interaction types and drug claims made around a gene being a target.

*InWeb*. InWeb_IM data[29] is publicly available at: https://www.intomics.com/inbio/map.html#downloads ('inBio_Map_core_2016_09_12.zip'). We include a scored human protein-protein interaction network named InWeb_InBioMap. From this

resource we retrieve scores regarding experimental and inferred interactions which were yielded as the validation degree of each link recorded in the original analysis. The data is subject to feature engineering eventually representing whether each gene's direct (1-hop) or indirect (2-hop) interactions are druggable based on the experimental and inferred interaction scores (see also "Network feature engineering" section in Methods).

*StringDB*. Another integrated data source regarding protein-protein interactions is STRING[31], a database of known and predicted links spanning more than 24 M proteins and 5 K organisms. In DrugnomeAI we use data related to physical (direct) and protein (indirect) associations which is subject to the same feature engineering applied on the InWeb data (related to protein based and physical based features, respectively). The dataset is publicly available at https://string-db.org.

*Reactome*. The Reactome Knowledgebase[43] is publicly available at https://reactome.org. We incorporate data from Reactome which provides information on molecules and their relations organised into biological pathways and processes. In order to enhance DrugnomeAI with additional information on protein-protein interactions, we capture all gene-level associations (as presence or absence of an association) from Reactome's full network representation and pre-process it as described in the "Network feature engineering" section of Methods.

*DGIdb*. The Drug-Gene Interaction Database (DGIdb)[28] constitutes a resource of information on drug-gene interactions and druggable genes gathered from literature, databases, and other web-based sources and is publicly available at www.dgidb.org. In DrugnomeAI we integrate information around the number of interaction types each gene has, thus capturing the count of drug-gene interactions per gene.

*CTDbase*. The CTD[44] is publicly available at http://ctdbase.org. We leverage information on chemical-gene interactions (CTD_chem_gene_ixns.csv.gz, CTD_chem_gene_ixn_types.csv) which are characterised by their degree and type. We process the information to retrieve the number of chemicals having certain interaction types with a gene. Due to the large number of interaction variants and the data sparsity for the least observed ones, we include only those types which are at or above the 50th percentile of the frequency distribution of all interaction types across the exome. Eventually, we use as features 65 chemical-gene interaction types, the number of unique interactions recorded per gene and the count of chemicals associated with interactions of the remaining variants (for those genes below the 50th percentile of the frequency distribution of interaction types). In addition, we integrate data regarding gene-pathway associations (CTD_genes_pathways.csv.gz), namely we select those pathways which are at or above the 90th percentile of their frequency distribution across the exome and assign a boolean flag representing whether a given pathway association exists for a gene. The selection criteria yielded 238 pathway associations that translate into 238 boolean features along with a count of other pathway associations observed for a gene.

*InterPro*. The InterPro database[45] is publicly available at https://www.ebi.ac.uk/interpro. We integrate information on classification of protein sequences into families, by assigning a boolean flag according to a gene's protein membership to a domain, family or super family, resulting in 97 features.

*OMIM*. Online Mendelian Inheritance in Man (OMIM) data is available subject to licensing at: https://www.omim.org. From this resource we use information on phenotype/disease associations with genes to extract the count of unique diseases per gene. In order to reduce the redundancy among the disease terms associated with each gene we employ a pre-trained nature language processing model, namely BioWordVec[46], to capture the semantic similarity between all pairs of disease terms. For every disease term, we first query all articles freely available at PubMed Central (PMC) with elastic search, returning the top 10 best matching documents per disease. We then extract a set of words within a 200-word window, centred around the matching disease term in each article extract and calculate their average embeddings, as provided by BioWordVec. The word embeddings serve for calculating semantic distances with their total distribution being used to infer the count of unique diseases associated with each gene. Specifically, the distinctness of disease terms is determined based on a one sample $t$-test performed on all pairwise disease distances, annotating a pair of diseases as distinct when $p < 0.05$.

**Annotation of druggable genes**. We employ two resources of documented drug targets as seed genes for the DrugnomeAI supervised learning framework (Table 1). The first one is Pharos (https://pharos.nih.gov/) which defines four tiers: *Tclin*, approved drugs with known mechanism of action; *Tchem*, genes having activities in ChEMBL or DrugCentral; *Tbio*, genes for which no known drug or small molecule activities exist but annotations of a Gene Ontology Molecular Function or Biological Process leaf term(s) with an Experimental Evidence code are recorded or else a confirmed OMIM phenotype can be found; and lastly *Tdark*, genes where virtually nothing around their druggability potential is known.

**Table 1 Resources for extracting labels around the druggability potential of genes, based on varying levels of supporting evidence.**

| Positively labelled data (known drug targets) | Category | Number of genes |
|---|---|---|
| Pharos Labels | Tclin | 610 |
| | Tchem | 1592 |
| | Tbio | 11,316 |
| Triage/Tier Labels | Tier 1 | 1411 |
| | Tier 2 | 658 |
| | Tier 3A | 845 |
| | Tier 3B | 1437 |

The second resource that served to determine gene druggability potential is the triage label system introduced in the druggable genome work[2]. Four tiers characterise the gene druggability potential as follows: *Tier 1*, genes being targets of approved small molecules and biotherapeutic drugs (as well as clinical-phase drug candidates); *Tier 2*, gene targets with known bioactive drug-like small-molecule binding partners as well as those with high sequence similarity with approved drug targets; and *Tier 3A*, secreted or extracellular proteins that have only distant similarity to approved drug targets, as well as *Tier 3B*, other members of key druggable gene families not already included in Tier 1 or 2.

**Data pre-processing**. DrugnomeAI combines data from a variety of gene-annotation sources being categorised into three groups according to their resulting features: druggability-specific, disease-specific, and generic resources (i.e. disease and/or tissue agnostic) that were originally included in mantis-ml. All the compiled data subjected to feature extraction result in almost 500 gene-associated attributes. Subsequently, DrugnomeAI applies automated pre-processing which comprises of removing highly correlated/redundant features and handling missing data. By default, when two features achieve a Pearson's correlation coefficient above 0.8 we retain only one of the highly correlated features. The same applies to the features whose missing data percentage exceeds a certain threshold (parameter "eda_-parameters -> missing_data_thres" in drugnome_ai/conf/.config; default value: 99% meaning that even sparse features having at least 1% non-missing data are considered). The remaining features having missing data are imputed with either a zero or the respective feature's median, depending on the biological context of each feature. Specifically, missing values in features representing a binary flag or a biologically relevant signal derived from computational or experimental studies performed only on a subset of genes, are imputed with zero. In the case of continuous variables, the choice of the median for imputation is governed by the fact that these attributes are obtained from research using different global reference sets of genes, therefore extrapolation of these values seems to be the most suitable strategy. Features related to protein networks are assigned a zero when no interaction exists. Finally, the data is standardised such that each variable has a zero mean and unit variance. In total, DrugnomeAI incorporates 324 features after pre-processing is complete. The full list of features is provided in Supplementary Data 9 along with detailed description for each of them in Supplementary Data 5 and in the DrugnomeAI portal ("Features/DrugnomeAI Feature Reference").

**Network feature engineering**. To further elucidate the druggability potential of genes we also integrate protein-protein interaction data which encompass common pathways or mappings of existing interactions between human proteins. To tap into the rich information captured in networks we pre-process all network/graph type of data to derive structured features that capture most of the information about the structure and connections of the original network. Specifically, for every gene we calculate the ratio of interactions with positively labelled (known druggable) genes to the total number of links associated with that gene (Supplementary Fig. 22). We first capture each protein's direct connections, i.e. those that are one edge apart in relation to the target of origin. We also expand the same logic to calculate the indirect connections that are two edges apart. As the number of edges apart increases, the number of associated connections (unlike the number of druggable pathways) grows exponentially, and thus, we limit the construction of this scoring metric for the first (one edge apart) and second (two edges apart) neighbours of each gene in the network.

**Boruta feature selection algorithm**. Boruta is an iterative feature selection method, on top of a Random Forest classifier, to determine if a feature has any statistically robust predictive power[27]. Features that are proved less statistically significant are eliminated. Unlike other feature selection methods where features are compared against each other, here, features are compared to "shadow" features. Shadow features are randomised versions of the original features. In a nutshell, a feature is considered significant if it is more significant than the most significant shadow feature. A basic Boruta iteration goes as follows:

1. Create shadow features and append them to the original features matrix
2. Train a random forest model and calculate the importance of original and shadow features
3. Identify *s'*: the most significant shadow feature
4. Identify hit features: all original features whose significance is higher than *s'*.

This iteration is repeated several times, then we obtain a binomial distribution of the number of times a feature is determined as a 'hit'. Three feature sets are eventually extracted from this distribution: (1) "Confirmed" features: these are features that are considered predictive and are located at the top (95th percentile) of the distribution, (2) "Rejected" features: these are features that are considered irrelevant and are located at the bottom (5th percentile) of the distribution, and (3) "Tentative" features: falling in the middle of the distribution (5th–95th percentiles) for which feature contribution is tentative.

**Validation data resources**

*Clinical evidence*. We employ the Open Targets Platform[8] (February 2021 release) for retrieving gene-level clinical evidence data. We extracted all genes supported by "known_drug" evidence and all clinical trials data including phase, indication and molecule type.

*Non-clinical evidence*. We have retrieved non-clinical evidence data that are available across the exome (genetic, pathways, somatic mutation, animal models, RNA expression and literature) from the Open Targets Platform (February 2021 release).

*PheWAS dataset*. We used an extended cohort of data from Wang et al.[23] which analysed nearly 450 K exomes from the UKB to identify relationships between rare protein-coding variants and 17,361 binary and 1419 quantitative phenotypes. For the enrichment analyses with top DrugnomeAI hits we focus on significant genes ($p < 5 \times 10^{-8}$) from PheWAS (Table 2).

**Comparison with other methods**. We compared DrugnomeAI with three published tools: (1) TargetDB[10], which employs a random forest model, (2) a hybrid deep learning model consisting of CNN, RNN with bi-directional long short term memory and deep neural networks (DNN) developed by Yu et al.[12], and (3) a decision tree-based meta classifier developed by Costa et al.[14]. TargetDB provides a pre-trained model, which we used to obtain the prediction scores for the 19,846 genes available in our dataset. Costa et al. provide druggability scores for 10,000 genes, which we used for the comparison analysis. Yu et al. provide the source code and the training dataset. We trained the hybrid model with the dictionary encoding of three feature sets (dipeptide composition, tripeptide composition and composition-transition-distribution) on their provided training data, and applied the model to predict the druggability of 19,846 genes in our dataset. We obtained amino-acid sequences from Uniport[42] and used the Protr[47] package in R for generating the features.

We used two data sources for comparisons: Open Targets tractability data (February 2021 release) for small molecules and antibodies and a list of genes with approved drugs from King et al.[35]. In the Open Targets tractability data, genes are grouped into several categories (also called buckets) of tractability depending on supporting evidence. Genes in Buckets 1–3 are supported by clinical evidence, while genes in the remaining buckets are supported by discovery precedence or have been predicted as druggable with varying levels of confidence (more details available at https://platform-docs.opentargets.org/target/tractability). To explore the models' behaviour in identifying genes with various strengths of supporting evidence, we assessed the models using genes in bucket 1, buckets 1–3, and remaining buckets as described in Table 3.

**Specialised DrugnomeAI models**. To train the therapeutic modality models, we retrieved genes targeted by small molecules or monoclonal antibodies from the Open Targets platform (February 2021 release) and PRTOACs targets published by Schneider et al.[38]. To generate the oncology and non-oncology models, we obtained the gene lists for CPD from the Open Targets platform for training CPD-sm, CPD-ab, non-CPD-sm, and non-CPD-ab models, while the cancer-sm gene list is obtained from the Cancer Therapeutics Response Portal (CTRP)[48]. Descriptions for each specialised model is available in Table 4.

**Integration of additional evidence from the open targets platform in the DrugnomeAI web resource**. We have integrated within the DrugnomeAI web resource additional types of evidence for each gene around their druggability profile (e.g. whether it has an approved drug, if it has been tested in clinical trials, etc.), its cellular localisation, the existence of homologues in other model organisms and its expression profile across multiple tissues. Even though some of these gene-level properties have been used as part of the labelled dataset and/or feature set of DrugnomeAI, we find it useful to accompany the DrugnomeAI results with this extra information to facilitate the interpretability of the results. All data were parsed from the Target dataset in the Open Targets platform downloads page (https://platform.opentargets.org/downloads), focusing on the following fields/

**Table 2 Significant PheWAS hits and overlap with top DrugnomeAI results.**

|  | Trait | Genes with $p$ values $< 5 \times 10^{-8a}$ | Overlap with top DrugnomeAI-Tclin[b] | Overlap with top DrugnomeAI-Tier1[b] |
|---|---|---|---|---|
| PheWAS | Binary | 210 | 31 | 33 |
|  | Quantitative | 508 | 73 | 82 |

[a]Top genes from PheWAS statistics (genes with $p$ value $< 5 \times 10^{-8}$).
[b]Overlap between top 5% ranked genes by DrugnomeAI-Tclin (or DrugnomeAI-Tier1) with top genes from PheWAS.

**Table 3 Validation datasets.**

| Dataset |  | Description | Number of genes |
|---|---|---|---|
| King et al. 2019 | – | Genes with approved drugs | 2201 |
| Open Targets (Small Molecules) | Bucket 1 | Genes with drugs in phase IV trials | 612 |
|  | Buckets 1–3 | Genes supported by data from clinical precedence | 895 |
|  | Buckets 1–8 | Genes supported by data from clinical or discovery precedence, or predicted tractable | 5034 |
| Open Targets (Antibody) | Bucket 1 | Genes with drugs in phase IV trials | 63 |
|  | Buckets 1–3 | Genes supported by data from clinical precedence | 331 |
|  | Buckets 1–9 | Genes supported by data from clinical precedence or predicted tractable | 9988 |

**Table 4 Datasets for specialised models.**

| Dataset | Description | Number of genes for training the model |
|---|---|---|
| Small Molecules (sm) | Genes with approved small molecule drugs or clinical-phase small molecule drug candidates | 885 |
| Antibody (ab) | Genes with approved monoclonal antibody drugs or clinical-phase monoclonal antibody drug candidates | 251 |
| PROTAC | Genes targeted by PROTAC drug modality supported by clinical evidence in buckets 1–3 (2 genes) literature evidence (267 genes) | 269 |
| CPD-sm | Genes with approved small molecule drugs or clinical-phase small molecule drug candidates for oncology diseases | 699 |
| CPD-ab | Genes with approved monoclonal antibody drugs or clinical-phase monoclonal antibody drug candidates for oncology diseases | 175 |
| Non-CPD-sm | Genes with approved small molecule drugs or clinical-phase small molecule drug candidates for non-oncology diseases | 186 |
| Non-CPD-ab | Genes with approved monoclonal antibody drugs or clinical-phase monoclonal antibody drug candidates for non-oncology diseases | 76 |
| Cancer-sm | Genes in cancer cell lines linked with small molecules from Cancer Therapeutics Response Portal (CTRP) | 322 |

columns: "subcellular locations", "homologues", "tractability" and "base-lineExpression". In more detail:

–  For subcellular locations, only the Uniprot source was used. Then locations were grouped into "membrane", "intracellular", and "secreted". "Secreted" was already defined as one of the localisation groups. For the "membrane" group we aggregated all localisations containing the word "membrane" (case insensitive) and not including the string "nucl" (to avoid the nuclear membrane localisations). The "intracellular" group was then defined by aggregating all remaining localisations.
–  To define homologues, we only considered the "orthologue_one2one" type.
–  For tractability, we extracted the following modality types: AB (antibody), SM (small molecule), while we also extracted Protein degradation (PR). In addition, we annotated whether the modalities fall in one of the following subcategories: "Approved Drug", "Advanced clinical", or "Phase 1 clinical".
–  For "baselineExpression", we extracted the "rna.z_score", "rna.level" and "protein.level" values from the corresponding tissues.

**Fine-tuning of gradient boosting hyperparameters**. Gradient Boosting (GB; implementation provided by the sklearn[49] Python module) outperformed all other classifiers (Random Forest, Extra Trees and Support Vector Classifier) that were tested as part of the supervised learning module of DrugnomeAI (Fig. 2a). We sought to fine-tune GB's default hyperparameters employed by DrugnomeAI, which were originally inherited from mantis-ml: n_estimators: 500; max_features:

sqrt, max_depth: 20, min_samples_leaf: 4, min_samples_split: 5, learning_rate: 0.1. We first optimised for "max depth", to reduce the search space of hyperparameters. Various "max depth" values were tested (1, 2, 3, 5) yielding best results (in terms of average AUC) with max depth = 5, which was retained as the default value for the rest of the analysis (we did not increase the parameter value any further to avoid extreme overfitting of the gradient boosting trees to the training set). We then performed gride-search for the "number of estimators": [10, 50, 100, 200, 300, 400, 500] and "learning rate": [0.001, 0.01, 0.1, 1] hyperparameters. The analysis was performed using two different gene-annotation options (either Tclin or Tier 1) to establish the consistency and robustness of the grid-search results across multiple labelled datasets. We eventually found that learning rate of 0.1 yields the best AUC results for both label sets. As for the "number of estimators" parameter, we observed that the AUC score plateaus at 200 estimators, with further increment of the estimators leading to overfitting and excessive skewness of the gene probabilities, thus selecting 200 as the default number for this parameter.

**Computational resource requirements**. DrugnomeAI was run on a high performance computing cluster requiring 200 GB memory and 4 cores provided by a pool of Intel processors (Broadwell, Skylake, CascadeLake) and AMD Rome. The running time was 2h40min and 3h12min for training DrugnomeAI-Tclin and DrugnomeAI-Tier1, respectively.

**Statistics and reproducibility**. The details about sample sizes, parameters and steps of statistical analysis are provided in relevant methods and results sections,

figure legends, and tables where applicable. All statistical analysis is performed in Python.

**Reporting summary**. Further information on research design is available in the Nature Portfolio Reporting Summary linked to this article.

## Data availability

Labelled gene lists from Pharos are available at: https://github.com/astrazeneca-cgr-publications/DrugnomeAI-release/drugnome_ai/data/PHAROS/pharos_GF_wINDEX.csv. Triage gene lists are available at: https://github.com/AstraZeneca-CGR/drugnomeAI/blob/master/drugnome_ai/data/labels/gene_druggable_labels.csv. Labelled gene lists for training specialised models are available at: https://github.com/astrazeneca-cgr-publications/DrugnomeAI-release/misc/gene_lists. The data for generating the graphs are provided in Supplementary Data and at the DrugnomeAI Github repository. The data used in this study have been obtained from the following resources: PHAROS (http://juniper.health.unm.edu/tcrd), InWeb (https://www.intomics.com/inbio/map.html#downloads), StringDB (https://string-db.org), Reactome (https://reactome.org), DGIdb (www.dgidb.org), CTDbase (http://ctdbase.org), InterPro (https://www.ebi.ac.uk/interpro), OMIM (https://www.omim.org), Open Targets Platform (https://platform.opentargets.org/downloads), and CTRP (https://portals.broadinstitute.org/ctrp.v2.1/?page=#ctd2Target). Additionally, we used datasets from the following publications: Finan et al. (https://www.ncbi.nlm.nih.gov/pmc/articles/PMC6321762/#SMtitle), King et al. (https://journals.plos.org/plosgenetics/article?id=10.1371/journal.pgen.1008489#sec018), Wang et al. (https://www.nature.com/articles/s41586-021-03855-y#data-availability), Schneider et al. (https://www.nature.com/articles/s41573-021-00245-x#Sec10), and Costa et al. (https://bmcgenomics.biomedcentral.com/articles/10.1186/1471-2164-11-S5-S9#additional-information).

## Code availability

The DrugnomeAI package and code for reproducing validation analysis, along with the training/validation datasets and instructions for installing and running the software are available in the GitHub repository: https://github.com/astrazeneca-cgr-publications/DrugnomeAI-release. The web application to visualise the DrugnomeAI predictions and the key features around gene druggability, per disease type and modality are available here: http://drugnomeai.public.cgr.astrazeneca.com.

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

## Acknowledgements

We thank Andrew Davis for his valuable feedback on this work.

## Author contributions

D.V., E.T. and J.S developed the DrugnomeAI models. A.R., E.T. and D.V. completed the model assessments and validation of predictions. L.M. developed the web portal with the DrugnomeAI results. R.D., A.R.H., P.H., O.E., S.P. and D.V. contributed to the organisation of the project. A.R., E.T. and D.V. wrote the paper.

## Competing interests

A.R., L.M., R.D, P.H., O.E., A.R.H., S.P. and D.V. are employees of AstraZeneca. A.R., R.D, P.H., O.E., A.R.H., S.P. and D.V. are shareholders of AstraZeneca. L.M.'s work was funded by the AstraZeneca post-doctorate program. E.T. and J.S. declare no competing interests.
