## [Peer Review File · Communications Biology]

Reviewers' comments:

Reviewer #1 (Remarks to the Author):

This paper expands upon previous attempts to use ML to predict druggable gene targets in several areas, including superior performance on benchmarking data, and predictions for novel modalities such as PROTACs. Furthermore, the web interface for DrugnomeAI is quite intuitive and should be easily queried by biologists interested in their favorite gene or hit from a screen. My main concern initially was that the data would be overfitted since the Tclin genes were used for training. The authors acknowledge this issue and claim that enrichment for OMIM genes and PheWAS loci alleviates this concern. However, this paper would benefit from additional exploration of the top predicted druggable genes as detailed below.

1) For the 239 and 387 top ranking genes from in Tclin and Tier1 predictions without clinical support provide illustrative comparisons of the number of genes with each level of support and fishers exact test for enrichment (e.g. 239/239 have 1 type of support, x/239 have 2 types of support, ... 24/239 have 6 types of support).

2) Are there any identifiable features of top ranking genes that have not been pursued clinically? Can some feature (expression levels, gene length, conservation with other primates etc.) predict why a gene has not yet been targeted?

Reviewer #2 (Remarks to the Author):

The authors of the manuscript developed a machine learning framework called DrugnomeAI to predict whether a protein-coding gene is druggable, thus estimating the ability to induce a therapeutic effect because of activation or inhibition of a gene by some therapeutic perturbation (e.g. small molecule, antibody, or proteolysis-targeting chimera). They integrated gene-level features (324 total features) from 15 sources and they utilized a Gradient Boosting classification model coupled with a previously published stochastic semi-supervised learning approach where an ensemble of classifiers is trained on different subsets of the data (split in a stratified manner to counter class imbalance) to estimate the druggability likelihood via a predicted average probability from all the classifiers. Additionally, they conducted a feature importance analysis to estimate the most influential gene-level features for the druggability prediction task, while also including different modalities of the framework for different therapeutic areas or perturbations.

The current study provides a valuable tool for the therapeutic design pipeline to identify whether a target of interest can be potentially perturbed with the therapeutic approach in mind, providing for the first time a framework for predicting gene druggability by proteolysis-targeting chimeras. Additionally, the authors' feature importance analysis provides new information about the usefulness of multiple gene features to estimate the druggability of a target, concluding also that systems biology-level features derived from protein-protein interaction networks are the most important predictors in estimating target druggability, meaning that their approach also considers the indirect effects on a target in the biological system. Finally, they sufficiently compared their framework with other published approaches showing that they achieved higher performance and hit rate with statistical significance, and also, they identified potential target genes not yet included in known drug-target studies.

Some specific comments/recommendations regarding the manuscript are:

Comment #1: No matter how repetitive or redundant, I believe a (even brief at least) description is needed in the method's section for the feature importance analysis and the Boruta algorithm used for that, as well as the score calculated and used to signify the importance of each predictor. The feature selection process is a big and important part of the manuscript and the reader would benefit from more details.

Comment #2: The authors of the manuscript claim that gradient boosting is the method performing the best of the methods they used, but p.values for that comparison should be

provided. I wonder if a statistically significant difference with random forest truly exists, especially since there is a big correlation between the two methods and the difference seems small (Figure 2A, Supplementary Fig. 1, Supplementary Fig. 14).

Comment #3: The approach is also compared with a published neural network-based approach but because of the difference in the approach, the neural nets are trained on different features. It would be nice to have also a simple neural net classifier trained on the same features and compared with the other traditional machine learning approaches.

Comment #4: I believe a few words for proteolysis-targeting chimeras (PROTACs) in the introduction or methods section are necessary especially since one of the novelties of the manuscript is related to druggability predictions for this specific therapeutic perturbation.

Comment #5: A distinction between druggability and the strength of the binding between a ligand and a target must exist in the introduction, as well as a more thorough explanation of what druggability is so that the reader to fully understand the possible applications of the model.

Comment #6: The first section of the results provides a general overview of the approach and the main results but it is too long. I believe that the feature selection process needs a section on each own to highlight to the reader the founding regarding important gene-level predictors.

Dear Editor and Reviewers:

We would like to thank the editor and reviewers for their comments. These have enabled us to improve the validation of our approach and extract further insights. Below, we address each of the specific reviewer comments and cite the location of the edits corresponding to the marked-up version of our resubmission.

Response to Reviewer #1

This paper expands upon previous attempts to use ML to predict druggable gene targets in several areas, including superior performance on benchmarking data, and predictions for novel modalities such as PROTACs. Furthermore, the web interface for DrugnomeAI is quite intuitive and should be easily queried by biologists interested in their favorite gene or hit from a screen. My main concern initially was that the data would be overfitted since the Tclin genes were used for training. The authors acknowledge this issue and claim that enrichment for OMIM genes and PheWAS loci alleviates this concern. However, this paper would benefit from additional exploration of the top predicted druggable genes as detailed below.

*We thank the reviewer for the comment and would like to elaborate on the diversity of training sets we have employed in DrugnomeAI. Overall, we have used seven datasets to develop the DrugnomeAI disease-agnostic models, defining druggable targets at various levels of stringency, as detailed in section “**Annotation of druggable genes**” (pages 22-23; **Table 1**). The Tclin dataset is the most stringent dataset as it consists of only approved drug targets and we agree that it offers a fairly strict and limited scope of druggability, since it depends on historically established drug targets. We do provide it though as one of the DrugnomeAI models in order to detect any strong patterns of druggability identified across the exome, based on the known approved drug targets. However, druggability within DrugnomeAI is not represented by a single model. Other training sets, such as Tier1 and Tchem, are less stringent since they are focusing on broader and more diverse aspects of druggability. Moreover, we used another eight datasets for training the domain-specific models: three drug-modality specific, three oncology and two non-oncology specific models, detailed in section “**Specialised DrugnomeAI models**” (pages 26-27; **Table 4**). The end goal is that we provide a holistic view of the druggability profile of each gene, as it is captured by 15 different models, each contributing to a different aspect of their druggability potential. Thus, the accompanying web app provides druggability scores from all the 15 models to provide a comprehensive estimate of genes druggability. We have added a sentence in “Introduction” (3rd paragraph, **page 2**) to clarify the concept of druggability inferred from DrugnomeAI.*

1) For the 239 and 387 top ranking genes from in Tclin and Tier1 predictions without clinical support provide illustrative comparisons of the number of genes with each level of support

and fishers exact test for enrichment (e.g. 239/239 have 1 type of support, x/239 have 2 types of support, ... 24/239 have 6 types of support).

We thank the reviewer for the suggestion. We now provide enrichment analyses for the top ranking genes from Tclin and Tier1 without clinical support, against one or more types of support accompanying each of them, performed via Fisher's exact test against multiple random subsets of genes (null subsets). For each enrichment analysis, we report the median p-value achieved across 10 iterations against random (null) genes sets (Figure 4D, Supplementary Fig. 27). We found all 239 (Tclin-based predictions) and 386 out of 387 (Tier1-based predictions) genes to be associated with diseases and supported by at least two types of non-clinical evidence (Tclin Fisher's exact $p=1.5 \times 10^{-08}$; Tier1 Fisher's exact $p=5.2 \times 10^{-10}$; Figure 4D, Supplementary Fig. 27). Large proportions of the Tier1 and Tclin top ranking genes without clinical evidence are further supported by three, four or even five types of support, and significantly so compared to random gene sets. Finally, there are 24 genes from Tclin-based predictions (Fisher's exact $p=2.4 \times 10^{-3}$) and 26 genes from Tier1-based predictions (Fisher's exact $p=1.3 \times 10^{-2}$) that are supported by six types of evidence (Figure 4D, Supplementary Fig. 9, 27). While all levels of support are statistically significant, we observe that for genes supported by six types of evidence, Fisher's exact test p-value are relatively lower compared to the other analyses. This is expected though due to the small number of genes supported by all six types of non-clinical evidence. Overall, it's notable that the top hits predicted by DrugnomeAI (without having prior clinical evidence) are highly and significantly enriched for several types of non-clinical evidence, suggesting that they are more likely to be biologically relevant for the problem of druggability. We report these results in the revised manuscript in section "Genes with no prior evidence in clinical development" (pages 10-11).

2) Are there any identifiable features of top ranking genes that have not been pursued clinically? Can some feature (expression levels, gene length, conservation with other primates etc.) predict why a gene has not yet been targeted?

We explored and now provide a section around the features of genes not previously pursued clinically to examine whether there are any identifiable traits that would distinguish them from genes selected for clinical development. We plot the kernel density estimate (KDE) of the top 20 features from Tclin and Tier1 and employ the Chi-squared statistical test to compare the distribution of any of these features between genes with or without clinical evidence (Supplementary Fig. 25-26). Top ranked features include monoclonal count, antibody count, protein sequence length, and DGldb interaction types ($p\text{-value} < 1.0 \times 10^{-308}$) where we observe that genes without clinical evidence have on average smaller values than genes that have been selected for clinical development. Notably, the CTD processes "decreases metabolic processing" and "increases uptake" have non-zero distributions among the genes without clinical evidence and are significantly different from the distributions of genes with clinical support. This may suggest that genes with no prior clinical evidence may be involved in metabolic pathways which are highly complex or more challenging to target.

*Other significant features include associated pathways and interactions from the Comparative Toxicogenomics Database (CTD). For example, we observe that “CTD increased cleavage” is highly present among genes that have been selected for clinical development but is depleted in genes without clinical evidence. That is expected as cleavage is one of the most established steps involved in drug mechanism of action, such as antibody drug conjugates for treating tumours, and seems to have already been studied and covered extensively among known drug targets. Detailed explanation of the features is available in **Supplementary Table 20**. We comment on these results in the revised manuscript section “**Genes with no prior evidence in clinical development**” (**pages 10-11**).*

Response to Reviewer #2:

The current study provides a valuable tool for the therapeutic design pipeline to identify whether a target of interest can be potentially perturbed with the therapeutic approach in mind, providing for the first time a framework for predicting gene druggability by proteolysis-targeting chimeras. Additionally, the authors’ feature importance analysis provides new information about the usefulness of multiple gene features to estimate the druggability of a target, concluding also that systems biology-level features derived from protein-protein interaction networks are the most important predictors in estimating target druggability, meaning that their approach also considers the indirect effects on a target in the biological system. Finally, they sufficiently compared their framework with other published approaches showing that they achieved higher performance and hit rate with statistical significance, and also, they identified potential target genes not yet included in known drug-target studies.

We thank the reviewer for their support on our current work and their suggestions to improve upon it.

Some specific comments/recommendations regarding the manuscript are:

Comment #1: No matter how repetitive or redundant, I believe a (even brief at least) description is needed in the method’s section for the feature importance analysis and the Boruta algorithm used for that, as well as the score calculated and used to signify the importance of each predictor. The feature selection process is a big and important part of the manuscript and the reader would benefit from more details.

*We thank the reviewer for the suggestion. We have now added a new section in Methods to provide an explanation of the Boruta algorithm in the revised manuscript in section “**Boruta***

feature selection algorithm” (page 24). We now also emphasise an ablation analysis on the optimal initial feature set employed by DrugnomeAI (pages 6-7).

Comment #2: The authors of the manuscript claim that gradient boosting is the method performing the best of the methods they used, but p-values for that comparison should be provided. I wonder if a statistically significant difference with random forest truly exists, especially since there is a big correlation between the two methods and the difference seems small (Figure 2A, Supplementary Fig. 1, Supplementary Fig. 14).

We now provide results from DeLong tests to compare the AUC scores attained by the Gradient Boosting against all other classifiers (Random Forest, Extra Trees, Support Vector Classifier and a newly introduced Deep Neural Net classifier) based on the Tclin and Tier1 labelled datasets. We observe that Gradient Boosting significantly outperforms all other classifiers for both the Tclin and Tier1 labelled datasets:

- **Tclin dataset** – DeLong test p-values of Gradient Boosting vs:
 - Random Forest: $p = 4.34 \times 10^{-18}$
 - Extra Trees: $p = 1.01 \times 10^{-18}$
 - SVC: $p = 2.44 \times 10^{-10}$
 - DNN: $p = 6.58 \times 10^{-12}$
- **Tier1 dataset** – DeLong test p-values of Gradient Boosting vs:
 - Random Forest: $p = 5.04 \times 10^{-29}$
 - Extra Trees: $p = 2.83 \times 10^{-30}$
 - SVC: $p = 3.32 \times 10^{-15}$
 - DNN: $p = 1.71 \times 10^{-10}$

The results are reported in the revised manuscript and in a new Supplementary Figure (page 7; Supplementary Fig 24).

Comment #3: The approach is also compared with a published neural network-based approach but because of the difference in the approach, the neural nets are trained on different features. It would be nice to have also a simple neural net classifier trained on the same features and compared with the other traditional machine learning approaches. We now provide a Deep Neural Net classifier (DNN) built into the DrugnomeAI framework. DrugnomeAI’s DNN model inherits its parameters from the respective classifier of mantis-ml (https://github.com/astrazeneca-cgr-publications/mantis-ml-release/blob/master/mantis_ml/modules/supervised_learn/classifiers/ensemble_lib.py), selected via grid search with 10-fold cross-validation. Specifically, the adopted DNN model has the following parameters: 'regl': 0.01, 'hidden_layer_nodes': [32, 32], 'add_dropout': True, 'dropout_ratio': 0.3, 'optimizer': 'Adagrad', 'epochs': 50, 'batch_size': 128.

We applied the DNN model on the Tclin and Tier datasets. DNN achieved AUC scores of 0.975 and 0.949 on Tclin and Tier1, respectively, though significantly underperforming the Gradient Boosting classifier achieving AUC scores of 0.990 (DeLong p-value = 6.58×10^{-12}) and 0.967 (DeLong p-value = 1.71×10^{-10}) on the same datasets. The results are reported in the revised manuscript (page 7, Supplementary Fig. 24).

Comment #4: I believe a few words for proteolysis-targeting chimeras (PROTACs) in the introduction or methods section are necessary especially since one of the novelties of the manuscript is related to druggability predictions for this specific therapeutic perturbation. We agree with the reviewer and thus we now provide a paragraph with an explanation of the PROTACs in the revised manuscript (within the "Introduction" section, page 3).

Comment #5: A distinction between druggability and the strength of the binding between a ligand and a target must exist in the introduction, as well as a more thorough explanation of what druggability is so that the reader to fully understand the possible applications of the model.

We have now added an explanation clarifying the scope of druggability and its difference with the concept of ligandability (highlighted new section within "Introduction", page 2).

Comment #6: The first section of the results provides a general overview of the approach and the main results, but it is too long. I believe that the feature selection process needs a section on each own to highlight to the reader the founding regarding important gene-level predictors.

We agree with the reviewer and have now moved the feature selection process to its own section in the revised manuscript entitled "Analysis of significant druggability-associated features with ablation and Boruta" (pages 6-7).

REVIEWERS' COMMENTS:

Reviewer #1 (Remarks to the Author):

I appreciate the effort the authors made addressing all of the reviewers comments. I am now satisfied that this manuscript is suitable for publication.

Reviewer #2 (Remarks to the Author):

I have no further comments. In the revised manuscript the authors have addressed all my concerns and comments from my original review.